# Reactive Extrusion and Magnesium (II) *N*-Heterocyclic Carbene Catalyst in Continuous PLA Production

**DOI:** 10.3390/polym11121987

**Published:** 2019-12-02

**Authors:** Rosica Mincheva, Satya Narayana Murthy Chilla, Richard Todd, Brieuc Guillerm, Julien De Winter, Pascal Gerbaux, Olivier Coulembier, Philippe Dubois, Jean-Marie Raquez

**Affiliations:** 1Rosica Mincheva, Satya Narayana Murthy Chilla, Richard Todd, Brieuc Guillerm, Olivier Coulembier, Philippe Dubois, Jean-Marie Raquez. Laboratory of Polymeric and Composite Materials, University of Mons, 23 Place du Parc, B-7000 Mons, Belgiumsatyamurthy.chilla@gmail.com (S.N.M.C.); richard.todd@umons.ac.be (R.T.); Brieuc.Guillerm@umons.ac.be (B.G.); Olivier.coulembier@umons.ac.be (O.C.); philippe.dubois@umons.ac.be (P.D.); 2Julien De Winter, Pascal Gerbaux. Organic Synthesis and Mass Spectrometry Laboratory, University of Mons, 23 Place du Parc, B-7000 Mons, Belgium; julien.dewinter@umons.ac.be (J.D.W.); pascal.gerbaux@umons.ac.be (P.G.)

**Keywords:** ring-opening polymerization, poly(lactide), reactive extrusion, carbene catalyst

## Abstract

Reactive extrusion and magnesium (II) *N*-heterocyclic carbene catalyst are successfully employed in continuous polylactide synthesis. The possibility of using six-membered *N*-heterocyclic carbene adducts to act as efficient catalysts towards the sustainable synthesis of poly(l-lactide) through ring-opening polymerization of l-lactide (LA) is first investigated in bulk batch reactions. Under optimized solvent-free conditions, polylactide (PLA) of moderate to high molecular weights and excellent optical activities are successfully achieved. These promising results are further applied in the continuous production of PLA in an extruder.

## 1. Introduction

Environmental concerns have pushed both scientists and industries towards the development of sustainable plastics, e.g., materials of an annually renewable origin, obtained via efficient synthetic procedures (low energy consumption and low components of toxicity (if any), solvent consumption, and waste) and biodegradability (or at least recyclability) [1,2]. Amongst all already studied biobased/biodegradable polymers, polylactide (PLA) most closely encompasses these prerequisites. Indeed, its synthesis via solvent-free (bulk or melt) ring-opening polymerization (ROP) of lactide (LA) allows for getting access to continuous processing via reactive extrusion (REx)^3^—a continuous process positioned as an alternative to batch reactors.

REx affords “more with less” in terms of consumption of raw materials, reagents, energy, and time, and also requires less investments. An extruder (mainly a co-rotating twin screw extruder) is employed, where the chemical reaction (bulk polymerization/depolymerization, chemical modification, saponication, and enzymatic reactions) is performed under elevated mixing at molecular level, and has already been proven to be effective for PLA production [3,4,5]. The only concern hindering it from being successful comes from the use of tin(II) derivatives as catalysts during REx [3,4]. The toxicity of metal traces remains a serious issue when PLA-based materials are implemented as biomedical devices [6,7]. A large variety of purely organic and transition metal-free catalysts have therefore been investigated for the last 20 years and found to be efficient in solution [8]. Nevertheless, their applicability in bulk, (where the polymerization temperature surpasses 150 °C) remains inadequate due to the significant incidence of undesirable side-reactions such as inter- and intra-molecular transesterification reactions as well as epimerization, yielding PLA-based materials of low crystallinity and poor thermomechanical performances [8,9]. Alternatively, the use of low-toxicity and low-cost metals such as zinc and magnesium-based catalysts seems promising for lactide polymerization toward ROP [10,11,12]. Amongst these catalysts, despite being rare to date, Mg complexes with N-heterocyclic carbenes (Mg–NHC) [13,14], especially with functionalized NHC ligands [15,16,17,18], displayed particularly short Mg–C bonds in the solid state, and exhibited interesting ligand deprotonation and reduction chemistry, forming an organic radical centered on the carbene ring in solution. However, to our knowledge, no studies have reported the use of Mg–NHCs in bulk conditions.

In regards to solely NHC carbenes, the less nucleophilic ones, i.e., six-membered N-heterocyclic carbenes, are more thermally stable and mild enough to activate lactide monomer in bulk under a relatively low polymerization temperature (a maximum of 130 °C). To shift their thermal stability towards polymerization temperature appropriate to the industrial production of PLA, i.e., above 170 °C, significant efforts have been made on electronic stabilizations through the use of suitable ligands, yielding labile NHC–based adducts [19,20,21,22]. Such adducts have shown to be able to liberate active free-carbene in other bulk polymerization systems and maintain their efficiency at high temperatures [8,23]. The literature-gathered knowledge prompted us to investigate 6-membered NHC Mg adducts for the bulk production of PLA in a continuous manner.

Three types of carbene-based adducts were initially investigated in bulk batch reactions, which allowed for optimizing reaction conditions and choosing the best performing catalyst. The transfer of the optimized solvent-free conditions towards the continuous ring-opening polymerization of l,l-lactide (l,l-LA) was then performed, and polylactide (PLA) of moderate to high molecular weights and preserved optical activities was successfully produced.

## 2. Experimental

### 2.1. Materials

l,l-lactide (l,l-LA, GALACTIC, Celles, Belgium) was recrystallized from dried toluene and stored in a glove box prior to use. CH_2_Cl_2_, toluene, and THF were dried by a solvent purification system (SPS, MBraun, Breukelen, The Netherlands) under N_2_ flow. KOtBu was purified by sublimation; all other starting materials and chemicals were purchased from Sigma-Aldrich (Overijse, Belgium) or VWR (Leuven, Belgium) and used without further purification. All air or water sensitive chemicals were stored in a glove-box under inert atmosphere prior to use. Unless mentioned, all reactions were carried out under a nitrogen atmosphere by a standard Schlenk technique.

### 2.2. Syntheses

#### Synthesis of the NHC

The synthesis of the NHC (Scheme 1) was performed without special precautions under air.

**1,3-dimethyl-3,4,5,6-tetrahydropyrimidin-1-ium tetrafluoroborate (1b)**. The synthesis of compound **1b** was performed according to the literature [22] via cyclisation of diamines. Briefly: *N,N′*-dimethyl-1,3-propanediamine (DMPA, 3.0 g, 1 eq), ammonium tetrafluoroborate (3.07 g, 1 eq) and trimethylorthoformate (2.46 g, 1.2 eq) were added to a round bottom flask and heated at 120 °C for two hours under stirring. The mixture was then filtered to yield clear oil and dried at 90 °C under reduced pressure overnight. Yield = 91 %. ^1^H NMR (500 MHz, DMSO-*d_6_*, δ): 8.18 (s, 1H, CH), 3.27 (t, 2H, CH_2_, J = 5.9 Hz), 3.11 (s, 3H, CH_3_), 2.76 (t, 2H, CH_2_, J = 5.6), 2.42 (s, 3H, CH_3_), 1.98–1.69 (q, 2H, CH_2_, J = 6.9).

**Isolated free-carbene (1)**. The dried **1b** (2.0 g, 1 eq) was further treated with potassium tert-butoxide (1.08 g, 1.5 eq) in 20 mL dry THF at r.t. in the glove-box under nitrogen atmosphere and the resulting mixture was filtered to yield the isolated free-carbene (**1**). Yield = 100 %. ^1^H NMR (500 MHz, DMSO-*d_6_*, δ): 3.34 (s, 1H, :CH), 2.14 (t, 4H, CH_2_, J = 5.4 Hz), 1.61 (s, 6H, CH_3_), 1.51 (q, 2H, CH_2_, J = 6.8).

**NHC-Mg complex (2)**. Freshly synthesized 1 (2.0 g, 1 eq) was dispersed in 100 mL dry THF, and added to a separately prepared dispersion of potassium tert-butoxide (1.68 g, 1.5 eq) in 10 mL THF and the blend dispersion was transferred to magnesium chloride (0.95 g, 1 eq) in a glove-box under anhydrous conditions. Upon prolonged stirring under inert atmosphere, a precipitate was formed, filtered, and dried under reduced pressure to yield **2** as an off-white powder. Yield = 54 %. ^1^H NMR (500 MHz, DMSO-*d_6_*, δ): 8.72 (s, 1H, :CH), 3.17 (t, 4H, CH_2_, J = 5.5 Hz), 1.99 (s, 6H, CH_3_), 1.13 (q, 2H, CH_2_, J = 6.5).

**NHC-carboxylate (3)**. Freshly prepared **1** (1 g, 1 eq) was dispersed in 10 mL dry THF. It was added to a separately prepared dispersion of potassium tert-butoxide (0.701 g, 1.25 eq) in 10 mL dry THF and then bubbled with carbon dioxide at 0°C for 1 h. Reaction was stopped by solvent evaporation and the formed sticky white precipitate was recrystallized from dry dichloromethane and dry diethyl ether. Yield = 90 %. ^1^H NMR (500 MHz, DMSO-*d_6_*, δ): 4.75 (s, 3H, CH_3_), 3.26-3.31 (t, 2H, CH_2_, J = 5.4 Hz), 3.21 (s, 3H, CH_3_), 3.08 (t, 2H, CH_2_, J = 5.8 Hz), 2.0 (q, 2H, CH_2_, J = 6.8).

**NHC-carbothioate (4)**. The synthesis of **4** was performed similarly to **3** but instead adding a blend dispersion of **1** (1 g, 1 eq) and potassium tert-butoxide (0.84 g, 1.5 eq) in 20 mL dry THF with CS_2_ (0.46 g, 1.2 eq). Yield = 96 %. ^1^H NMR (500 MHz, DMSO-*d_6_*, δ): 4.21 (s, 1H, CH), 3.36 (t, 2H, CH_2_, J = 5.9 Hz), 3.27 (t, 2H, CH_2_, J = 5.4 Hz), 3.04 (s, 6H, CH_3_), 1.91 (q, 2H, CH_2_, J = 7.1).

### 2.3. Batch Polymerization

In a typical polymerization experiment, l,l-LA (1 g, 0.7 mmol), and **2** (4 mg, 1.73 × 10^−5^ mol) are charged into a polymerization tube equipped with a three-way stop-cock and a rubber septum in the glove-box under controlled atmosphere. The sealed polymerization tube was removed from the glovebox and introduced into preheated oven at a pre-set temperature (150, 160, 170, 180, 190, or 210 °C). After a required reaction time, the polymerization was stopped by rapid quenching. The crude product was then dissolved in chloroform and precipitated in ethanol. The off-white polymer was isolated by filtration and dried at 60 °C under reduced pressure overnight. Conversion is calculated from SEC analyses of the crude product, while molecular characteristics are calculated from SEC of the purified polymer. To ensure identical conditions, for each experiment, the same type of vessel and stirrer was used, as well as the same stirring rate.

### 2.4. Reactive Extrusion

REx was tested into a 15 cm^3^ twin-screw DSM microcompounder (Xplore Instruments BV, Sittard, The Netherlands) at 170 °C and 75 rpm under nitrogen flow. The initial molar ratio [l,l-LA]_0_/[**2**]_0_ was fixed to 400. Premixing l,l-LA and **2** was performed at 130 °C for 1 min. Reaction progress was monitored by torque increase and followed up until constant values were obtained. The resulting product was finally recovered at 100 rpm and purified as described before.

## 3. Characterizations

^1^H NMR spectra were recorded at 25 °C in DMSO-*d_6_* or in CDCl_3_ on Bruker AMX-500 NMR spectrometer (Leiderdorp, The Netherlands). Chemical shift scale was referenced using TMS as an internal standard.

Size-exclusion chromatography (SEC) was performed in chloroform (CHCl_3_) at 30 °C using an Agilent liquid chromatograph equipped with an Agilent degasser (Diegem, Belgium), an isocratic HPLC pump (flow rate = 1 mL min^−1^), an Agilent autosampler (loop volume = 100 μL; solution concentration = 2 mg mL^−1^), an Agilent-DRI refractive index detector (Diegem, Belgium), and three columns: a PL gel 5 μm guard column and two PL gel Mixed-B 5 μm columns (linear columns for separation of *M*_w_ (PS) ranging from 200 to 4 × 105 g mol^−1^). Polystyrene standards were used for calibration.

MALDI-ToF mass spectra were recorded using a Waters Q-Tof Premier mass spectrometer (Zellik, Belgium) equipped with a nitrogen laser, operating at 337 nm with a maximum output of 500 J m^−2^ delivered to the sample in 4 ns pulses at a 20 Hz repeating rate. Time-of-flight mass analysis were performed in the reflectron mode at a resolution of about 10,000. The matrix-trans-2-[3-(4-tert-butylphenyl)-2-methyl-2-propenylidene] malononitrile (DCTB)—prepared as 10 mg mL^−1^ solution in acetone. The matrix solutions (1 μL) were applied to a stainless-steel target and air dried. Polymer samples were dissolved in dichloromethane to obtain 1 mg mL^−1^ solutions. Precisely 1 μL aliquots of these solutions were applied onto the target area already bearing the matrix crystals, and were then air dried. Finally, 1 μL of a solution of NaI (2 mg mL^−1^ in acetonitrile: water (1:1, *v*/*v*)) was applied onto the target plate. For the recording of the single-stage MALDI-MS spectra, the quadrupole (rf-only mode) was set to pass ions from m/z 1000 to 10000, and all ions were transmitted into the pusher region of the time-of-flight analyzer where they were mass analyzed with 1s integration time. Temperature: 80 °C and desolvation temperature: 120 °C. Dry nitrogen was used as the ESI gas. The quadrupole was set to pass ions from m/z 100 to 7000 and all ions were transmitted into the pusher region of the time-of-flight analyzer where they were mass-analyzed with a 1 sec integration time. Data were acquired in a continuum mode until acceptable averaged data were obtained. 

Differential scanning calorimetry (DSC) analyses were carried out on a DSC Q200 apparatus from T.A. Instruments (Zellik, Belgium), calibrated with indium. Instruments under nitrogen flow (heating and cooling rate 10 °C min^−1^) from r.t. to 200 °C. Samples (weight: about 5−7 mg) were sealed in aluminum pans and placed in the DSC cell.

## 4. Results and Discussion

As mentioned in the Introduction, the literature gathered knowledge prompted us to investigate 6-membered NHC (Scheme 1) for the bulk production of PLA in a continuous manner.

Three types of carbene-based adducts were investigated, i.e., the MgCl_2_–, CO_2_– and CS_2_-based adducts (**2**, **3** and **4** in Figure 1A) from the isolated free-carbene **1**. The motivation to select these ligands and corresponding adducts is based on their ability to efficiently stabilize NHCs with a special emphasis on magnesium chloride for its excellent biocompatibility and non-toxicity in biomedical applications.^24^ The initial analogue used for the synthesis of these adducts is constructed over cyclic amidinium halide salts. The 1,3-dimethyl-4,5,6-tetrahydropyrimidin-1-ium tetrafluoro-borate salt **1b** (Figure 1A) was then synthesized via cyclisation of diamines, in situ transformed in free-carbene by treatment with potassium tert-butoxide and lastly converted into zwitterionic (CO_2_, CS_2_) or magnesium adduct by appropriate treatment [23,24,25]. As such, the catalytic activity of adducts **2**, **3,** and **4** was evaluated in the bulk LA ROP at 150 °C (Table 1). Polymerization experiments were performed for an initial monomer-to-adduct molar ratio of 400 and at different reaction times. As references, reactivities of both free-carbene **1** and MgCl_2_ were performed under similar experimental conditions.

While MgCl_2_ and non-stabilized adduct **4** did not lead to any traceable polymerization extent, both free-carbene 1 and adduct 3 were efficient in terms of monomer conversion, but merely resulted into only low molar mass PLA samples (Table 1, ${\bar{M}}_{n}^{SEC}$ ≤ 8.100 g mol^−1^). Moreover, lactide epimerization at a degree as high as 38 % was recorded (as deduced from the presence of meso-lactide resonances at δ = 1.7 ppm), therefore excluding **1** and **3** for use in solvent-free conditions. Interestingly, the adduct **2** (Entry 2, Table 1) gave a higher conversion at a relatively high rate (92% conversion in 30 min), with nearly non-epimerized PLA (~5 %) of a number-average molar mass of 32.000 g mol^−1^.

Further polymerizations were then carried out at different time intervals by using **2** in sealed vessels at 170 or 190 °C ([l,l-LA]_0_/[**2**]_0_ = 400, mol/mol; Table 2). As already demonstrated for other organic catalytic systems, increasing the temperature gives contradictory results where faster kinetics are detrimental to the overall control of the process. As an example, a temperature increase from 170 to 190 °C results in sharp increase in the monomer conversion from 64% to72 % after only 5 min of reaction, but the molar mass decreases from 39,000 to 33,000 g mol^−1^. Above 190 °C, intensive transesterification reactions occurred in a similar way to for organometallic catalysts such as tin(II) octoate [4].

Following all experimental temperatures investigated here, working at 170 °C represents the best optimal conditions in terms of kinetics and control over the reaction. Considering that control experiments demonstrated that MgCl_2_ or free NHC **1** could not induce reasonable polymerizations (Table 1), a dual catalytic mechanism supporting the very high activity of **2** is proposed here. The concept of dual catalysis, where a Lewis acid supports a nucleophile to gain an increased catalytic effect, is not new and has already been reported several times in organic chemistry [26,27] and in polymerization catalysis [27,28]. NHC-adducts with MgCl_2_, serving as latent catalysts, have already been reported by Dove for the polymerization of lactones at high temperatures [28]. Inspired by their work, a cooperative action mechanism involving the dissociation of complex **2** at high temperature is here proposed (Figure 1B). Interestingly MALDI-ToF mass spectrometry evidenced the presence of both cyclic and α-hydroxy-ω-carboxy-linear PLA species (Appendix A). Such an observation tends then to indicate that during the reaction, the lactide monomer is activated by the MgCl_2_, and that the free NHC **1** can act both as a base (Figure 1B, route I) and a nucleophile (Figure 1B, route II). Therefore, the route I generates linear chains initiated by water impurities via a cooperative catalysis, while route II essentially leads to cyclic structures through a cascade catalysis.

Transesterification side reactions are clearly evidenced in Appendix A. Indeed, a 72-mass unit (u) separates each distribution signal instead the expected 144 u between consecutive ions. ^1^H NMR (Appendix A) clearly evidenced the most striking feature, which was the low extent of epimerization (as deduced from the presence of meso-lactide) for all investigated polymerization temperatures (Table 2), thus indicating that the Mg-protected carbene **2** is an efficient eco-friendly catalytic adduct for the bulk ROP of l-LA.

To further highlight the low extent of epimerization, four-cycle DSC analyses were carried out on a high molecular weight PLA obtained at 170 °C (Appendix A). As expected, the crystalline features are maintained after 4 cycles, showing a melting temperature at circa 167 °C characterized by a melting endotherm of about 32 J g^−1^. In contrast to other organic catalysts [29,30,31,32], this maintained constant of the melting endotherm along all 4 cycles suggests that the polymer-trapped adduct **2** does not interfere with PLA thermal properties.

These promising results led us to investigate the synthesis of PLA in a continuous manner using a microcompounder at 170 °C and at a rotation speed of 75 rpm under a nitrogen flow ([l,l-LA]_0_/[**2**]_0_ = 400, mol/mol). The efficiency of the adduct **2** was evaluated by first feeding the LA monomer into the microcompounder machine at 130 °C before increasing the extrusion temperature to 170 °C. Interestingly, machine force increased with the extrusion time, indicating the successful building up of PLA molecular weight. This further attests to the good efficiency of this carbene-type catalyst in promoting the continuous synthesis of PLA, and was supported by molecular characterizations (by both SEC and ^1^H NMR) giving a monomer conversion of 80 % together with *M*_n_ = 47.200 g mol^−1^. Once again, no significant epimerization was observed by ^1^H NMR spectroscopy (Figure 2A). The absence of epimerization was also confirmed by a five-cycle DSC analyses (Figure 2B) that were carried out on the resulting PLLA after purification (solubilization in chloroform and precipitation from MeOH). The as obtained results suggest that the carbene-MgCl_2_ catalyst is active and efficient in the synthesis of high molecular weight PLA via REx.

## 5. Conclusions

In conclusion, we developed an eco-friendly Mg-based six membered catalyst, which provided an effective and efficient polymerization route for the continuous synthesis of PLLA under solvent-free melt conditions. The resulting PLLA materials present controlled molecular weights, relatively low dispersities, and high optical purity. Another advantage of our new organocatalyst is the possibility to extend its efficiency in the continuous polymerization of LA, as carried out in a microcompounder. This enables us to consider this catalyst for the continuous synthesis of eco-friendly polylactide using reactive extrusion.

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
