# Peer review of "Reactive Extrusion and Magnesium (II) N-Heterocyclic Carbene Catalyst in Continuous PLA Production"

_polymers, 2019, doi:10.3390/polym11121987_

Round 1

Reviewer 1 Report

In this study, the authors developed Mg-based six-membered catalyst for the potential synthesis of PLLA. In brief, the article is interesting and can be considered for publication after a revision process. 

1) The characterization of the prepared sample is not enough. It is highly advised to perform FTIR and XPS study of their samples + add the corresponding description to the manuscript. 

2) Bench storage stability + PLLA production stability at longer period of time (larger than 60 min) should be reported.    

Author Response

First of all, Authors would like to thank the Editor and the Reviewers for their time and concern. Here below, we enclose our point-to-point answers to Reviewers’ comments.

Response: The structure of all products synthesized in this work was followed by NMR so to confirm the exact chemistry and ensure the low degree/the absence of side reactions (epimerization during REx). No chemical modification that could be followed by FTIR or by XPS was performed, and the possible residual presence of catalysts (in the purified samples) was not high enough to be followed/detected by FTIR and or XPS. Therefore, no FTIR or XPS are given in the manuscript.

Response: We thank the Reviewer for these interesting questions. Considering the polyester nature of the PLAs, bench storage stability was not even tested. Concerning the production stability at times overcoming 60 min, REx was limited to times avoiding any possible thermal degradation reactions.

Reviewer 2 Report

In this manuscript, reactive extrusion and magnesium (II) N-heterocyclic carbene catalyst have been successfully employed in continuous polylactide synthesis. The six-membered N-heterocyclic carbene adducts to act as efficient catalysts towards the sustainable synthesis of poly(L-lactide) through ring-opening polymerization of L-lactide (LA). It was first investigated in bulk batch reactions. Under optimized solvent-free conditions, polylactide (PLA) of moderate to high molecular weights and excellent optical activities were successfully achieved. These promising results are further applied in the continuous production of PLA in an extruder. It can be accepted for publication after a minor revision.

Additional comments

For proton NMR, please provide coupling constants for all d, t, q peaks. To verify the structures of prepared intermediates (catalysts), please also provide their 13C NMR data. ee is generally defined as enantiomeric excess. Thus, it would be better to define epimerization degree as ed.

Author Response

First of all, Authors would like to thank the Editor and the Reviewers for their time and concern. Here below, we enclose our point-to-point answers to Reviewers’ comments.

1. For proton NMR, please provide coupling constants for all d, t, q peaks.

Response: Coupling constants for d, t, and q protons of the NHCs are now listed in the revised manuscript (highlighted in yellow in the Exp.Part)

2. To verify the structures of prepared intermediates (catalysts), please also provide their 13C NMR data.

Response: As six-membered NHC are not new and their synthesis was adopted from the literature (ref. 23), 13C NMR spectra were not performed. For example, for compound 2, the chemical shifts are expected to be positioned at δ (ppm) = 49.3 (2C, =N-CH2), 37.3 ppm (2C, =N-CH3) and 23.1 ppm (1C, -CH2-CH2-)

3. ee is generally defined as enantiomeric excess. Thus, it would be better to define epimerization degree as ed.

Response: Authors thank the Reviewer for this correction. The “ee” is now defined as “ed” throughout the revised manuscript (highlighted in yellow throughout the manuscript).

Round 2

Reviewer 1 Report

no more comments